# The Association of Pachydrusen Characteristics with Choroidal Thickness and Patient’s Age in Polypoidal Choroidal Vasculopathy versus Central Serous Chorioretinopathy

**DOI:** 10.3390/ijms23158353

**Published:** 2022-07-28

**Authors:** Young Ho Kim, Yoo-Ri Chung, Chungwoon Kim, Kihwang Lee, Won Ki Lee

**Affiliations:** 1Department of Ophthalmology, Korea University College of Medicine, Seoul 02841, Korea; kimyh54067@naver.com; 2Department of Ophthalmology, Ajou University School of Medicine, Suwon 16499, Korea; cyr216@hanmail.net (Y.-R.C.); chungwoon92@aumc.ac.kr (C.K.); 3Retina Center, Nune Eye Hospital, Seoul 06198, Korea; wklee0921@gmail.com

**Keywords:** central serous chorioretinopathy, choroid, pachydrusen, polypoidal choroidal vasculopathy

## Abstract

We investigated the relationship between pachydrusen and choroidal thickness and age in eyes with polypoidal choroidal vasculopathy (PCV) and fellow eyes, compared to eyes with central serous chorioretinopathy (CSC). This retrospective study included 89 eyes with PCV and 146 eyes with CSC. The number, location, and shape of the pachydrusen and their association with choroidal thickness and age were analyzed. PCV eyes showed pachydrusen more frequently than eyes with CSC (52% vs. 20%, *p* < 0.001). Large solitary type and clustered type were more frequent in PCV eyes compared to CSC eyes (*p* = 0.003 and *p* = 0.001, respectively). Subfoveal choroidal thickness was associated with pachydrusen in eyes with PCV (odds ratio [OR] 1.006, 95% confidence interval [CI] 1.001–1.011, *p* = 0.027), while age was associated with pachydrusen in CSC eyes (OR 1.137, 95% CI, 1.073–1.205; *p* < 0.001). Pachydrusen were localized directly over the pachyvessel on optical coherence tomographic findings in approximately two thirds of PCV eyes and fellow eyes (62% and 67%, respectively). Risk factors for pachydrusen differ according to diseases. The presence of pachydrusen was associated with choroidal thickness in PCV, while the association with age was more prominent in CSC.

## 1. Introduction

The term “pachychoroid” indicates abnormally increased choroidal thickness often associated with dilated choroidal vessels [1]. Pachychoroid spectrum diseases have been described along with the advances of imaging modalities, and they include central serous chorioretinopathy (CSC), pachychoroid pigment epitheliopathy, pachychoroid neovasculopathy, polypoidal choroidal vasculopathy (PCV), and more recently focal choroidal excavation and peripapillary pachychoroid syndrome [1]. They commonly express focal or diffuse increases in choroidal thickness, but they are also associated with pachyvessels, that is, pathologically dilated choroidal vessels within the deep choroid, which causes compression of the choriocapillaris and eventually results in morphological and functional changes in the retinal pigment epithelium (RPE) [1].

Pachydrusen, termed pachychoroid-associated drusen, shows different patterns in appearance, grouping, and pattern of distribution [2]. Pachydrusen was first introduced by Spaide [2] as a new form of drusen associated with thicker choroid in non-exudative age-related macular degeneration (AMD). Since this report, its association with pachychoroid spectrum diseases has been investigated in several studies. Lee and Byeon [3] reported that pachydrusen were noted in almost half of PCV eyes and were associated with a thicker choroid compared to a typical soft drusen. Matsumoto et al. [4] reported that pachydrusen were more likely to be associated with age than with choroidal thickness. Previous studies on pachydrusen have focused on their manifestations limited by specific diseases, such as AMD, PCV, or CSC, respectively [3,4,5,6,7].

PCV and CSC are both considered to be pachychoroid spectrum diseases, and there have also been reports associating CSC with PCV [8], while their clinical manifestations differ. For age, CSC occurs mainly in young and middle-aged populations, while PCV involved aged people [1]. For choroidal thickness, various studies revealed that choroidal thickness differed by disease entities among pachychoroid spectrum diseases [9,10,11]. Accordingly, we investigated whether pachydrusen is associated with the aging process or with choroidal thickness in PCV eyes, compared to CSC eyes showing globally diffuse choroidal thickening.

## 2. Results

A total of 235 patients (78% male) were finally included in this study, and their mean age was 54.3 ± 14.5 years. We included 89 patients diagnosed with PCV and 146 patients with CSC, and their baseline characteristics are summarized in Table 1. Patients with PCV were significantly older, and they had more comorbidities, such as diabetes and hypertension, than patients with CSC.

Pachydrusen and all its subtypes were significantly more commonly found in eyes with PCV than in eyes with CSC (52% vs. 20%, *p* < 0.001, Table 1). The number of pachydrusen was also significantly higher in eyes with PCV than in eyes with CSC (1.1 ± 1.6 vs. 0.3 ± 0.8, *p* < 0.001, Table 1). Mean subfoveal choroidal thickness (SFCT) was significantly thinner in PCV eyes compared to CSC eyes (265.6 ± 105.1 μm vs. 402.3 ± 104.1 μm, *p* < 0.001, Table 1). The SFCT could not be measured in three eyes with PCV due to severe hemorrhage and/or exudation, which were among the undetermined groups for pachydrusen. The intraclass correlation coefficient (ICC) between examiners for SFCT was 0.919 (*p* < 0.001). Representative cases of CSC eyes and PCV eyes are presented in Figure 1 and Figure 2, respectively.

We performed logistic regression analysis to identify whether aging or SFCT was associated with pachydrusen. The results showed that SFCT was associated with a significant risk of pachydrusen in eyes with PCV (odds ratio [OR] = 1.005, 95% confidence interval [CI] 1.000–1.009, *p* = 0.037), while age was not associated (OR 0.997, 95% CI 0.946–1.050, *p* = 0.903). In contrast, age was associated with a significant risk of pachydrusen in CSC eyes (OR = 1.137, 95% CI 1.073–1.205, *p* < 0.001), while SFCT was not associated (OR 0.998, 95% CI 0.995–1.002; *p* = 0.342).

There were more pachydrusens in PCV eyes with a thick choroid than in those with a thin choroid (61% vs. 36%, *p* = 0.035, Table 2). The number of pachydrusen was also significantly higher in eyes with PCV with a thick choroid. According to age stratification, this was more prominent in the 60s; 79% of the eyes in the thick choroid group presented pachydrusen compared to 33% of the eyes in the thin choroid group (Table 2). The number of pachydrusen was also higher in the thick choroid group in the 60s, while there was no difference in other age groups. By location, peripapillary and perifoveal pachydrusens were more frequent in PCV eye with a thick choroid (Table 3). We also investigated the presence of pachydrusen in the fellow eyes of patients diagnosed with PCV and CSC using the McNemar test. There were no significant differences in the presence of pachydrusen between the study eyes and the fellow eyes of PCV (*p* = 0.216), as well as those of CSC (*p* = 0.186). The ICC between examiners for counting pachydrusens was 0.912 (*p* < 0.001).

Furthermore, we investigated the direct association between pachydrusen and pachyvessels in PCV. Of the total pachydrusen in PCV eyes, 43% were excluded because their association with pachyvessels could not be determined on optical coherence tomography (OCT) due to either poor image quality of the choroid or the location being out of range of OCT scans. Among detectable pachydrusen on OCT, approximately 62% of pachydrusen were localized directly over pachyvessels on OCT findings (Table 4). This was similar in the subgroup analysis stratified by SFCT, with 62% in both the thick and the thin choroid groups. Similarly, 67% of the pachydrusen in the fellow eyes were localized over the pachyvessel.

## 3. Discussion

This study showed that pachydrusen were more frequently present in eyes with PCV than in eyes with CSC. Moreover, the presence of pachydrusen was associated with choroidal thickness in PCV, while the association with age was more prominent in CSC. Considering the location of pachydrusen, probable ischemia of choriocapillaris via pachyvessels may play a role in this difference. Micro-ischemic changes to the choriocapillaris were suggested in PCV eyes in which pachydrusen was frequently identified [12]. Choriocapillaris flow deficits are also noted in CSC [13], while pachydrusens need time to be formed under RPE. We suppose that this may be associated with more pachydrusens in PCV compared to CSC.

After the introduction of EDI-OCT, choroidal thickness has been of interest in various chorioretinal disorders. In eyes with pachychoroid spectrum diseases, diffuse or focal thickening of the choroid was noted, usually associated with the vascular enlargement of Haller’s layer and relatively compressed choriocapillaris [14]. In this study, pachydrusen was more frequent in eyes with PCV than in eyes with CSC, and the mean number of pachydrusen was also greater. Similarly, studies performed in Indian cohorts reported a lower incidence of pachydrusen in CSC eyes than in PCV eyes [5,6]. In detail, 20% of CSC eyes presented pachydrusen in our study, and these results were similar to the 27% reported in a previous study with Japanese patients [4]. For PCV, pachydrusen were reported to account 47–49% in PCV eyes according to studies with Korean patients [3,15], which were similar to the 52% reported in this study.

CSC is now known to be associated with a diffusely thick choroid resulting from choroidal vascular dilatation, although the exact pathogenesis needs to be clarified [16,17,18]. In recent studies, upregulated cytokines including interleukin-6 and the presence of anti-endothelial cell antibodies in acute and chronic CSC suggest that inflammation and endothelial damage in choroidal vessels might be involved in the pathogenesis of CSC [19,20]. As the choroid is usually diffusely thickened in most CSC cases, the association of pachydrusen with choroidal thickness or pachyvessels may not be evident, as shown in this study. This was similar to previous results that reported no differences in choroidal thickness between CSC eyes with pachydrusen and those without pachydrusen [5]. However, it should be noted that age was associated with a significant risk of pachydrusen in CSC eyes, as Matsumoto et al. [4] reported. Age is a well-known factor that affects the choriocapillaris [11]. Our explanation for this finding concurs with their hypothesis that long-standing focal attenuation of the choriocapillaris and Sattler’s layer overlying pachyvessels might lead to the generation of pachydrusen, since pachydrusen was frequently localized within the delayed choriocapillaris filling site and over dilated outer choroidal vessels on indocyanine green angiography (ICGA), that is, the pachyvessels [4,21]. On OCT angiography (OCTA), choriocapillaris flow deficits were also noted in CSC, presented as a greater flow signal void area, both in affected eyes and fellow eyes, and 89% of the signal void area colocalized with pachyvessels [13,17]. Moreover, the decreased hydraulic conductivity and increased resistivity of Bruch’s membrane with aging would generate entrapment of drusen-associated proteins. These drusen-associated proteins along with blood protein and inflammatory cytokines from delayed choriocapillaris flow might be risk factors for the occurrence of pachydrusen in CSC patients.

In contrast, pachydrusen in eyes with PCV was associated with choroidal thickness in this study. The risk of pachydrusen increased by 1.005 times when the choroidal thickness was increased by 1 μm, while it did not present a significant association with age in PCV eyes. Interestingly, Jordan-Yu et al. [22] reported total lesions areas and polypoidal lesion areas of PCV tend to be larger in eyes with a thicker choroid. Although the relationship between pachydrusen and the pathogenesis of PCV remain poorly understood, these findings suggest a thicker choroid might be related to a more severe form of choroidal ischemia, influencing both the occurrence of pachydrusen and the phenotype of the PCV complex. However, we do acknowledge the variability of choroidal thickness in PCV eyes. Choroidal thickness in PCV is found to be very wide ranging from 42–649 μm [23]. This inhomogeneity of choroidal morphology can be attributed to many factors, such as age, age-related comorbidity, and chronic increased venous pressure. Further studies on intervortex venous anastomoses, choriocapillary ischemia, and choroid analysis using wide field angiography may increase our understanding of the relationship between the choroid thickness and pachydrusen.

It should be noted that pachydrusen was also found in 36% of PCV eyes with a relatively thin choroid in our study, and 62% of pachydrusen were localized directly over pachyvessels on OCT findings in both the thick and the thin choroid groups. The reason for a similar percentage of pachydrusen directly over pachyvessels whether the choroid is thick or thin might be related to the relative thickness of the choriocapillaris and Haller’s layer. This suggests that pachydrusen are associated with choriocapillary abnormalities in addition to choroidal thickness. These eyes showed an attenuated choriocapillaris layer, as shown in the representative case, due to pachyvessels occupying the full thickness of the choroid. Furthermore, even though the present study did not evaluate wide field ICGA of PCV patients, the reason for the frequent location of perifoveal pachydrusens in PCV eyes with a thick choroid might be associated with intervortex venous anastomoses and dilated choroidal veins at the watershed zone, as Spaide et al. suggested [24].

When investigating the exact correlation between pachyvessel and pachydrusen locations, our study revealed that 62% of pachydrusens showed underlying pachyvessels, which was less than the rate of 90% reported by Baek et al. [25] and more than the rate of 49.3% reported by Lee and Byeon [3]. This might be associated with the different identification modalities between studies, as Baek et al. [25] additionally used en face OCT images to verify the relationship between pachyvessels and pachydrusen. On the other hand, Lee and Byeon [3] simply calculated the percentage of eyes presenting pachyvessels among those with pachydrusen, without investigating exact matching locations on OCT. In this study, the presence of pachyvessels lying below the pachydrusen might not have been properly identified in some cases due to the scan interval of 125 μm of the OCT used.

The exact pathogenesis for pachydrusen remains unknown, but the following factors should be considered between PCV and CSC: the common abnormality, pachyvessels, and the transport role of Bruch’s membrane with aging. Pachydrusen might occur under RPE because of sustained hypoxia of the choriocapillaris and age-related damage to the Bruch-RPE complex, whether choroidal thickening is focal or diffuse or whether the choroid is thick or thin. As shown in this study, depending on the specific disease entity in the pachychoroid spectrum, the risk factors for pachydrusen might be different, such as thickness as in PCV or age as in CSC, since the diseases have their own clinical characteristics. Nevertheless, the cause of pachydrusen might be strongly associated with choriocapillaris insufficiency, followed by the thickened Haller layer and age-related changes in Bruch’s membrane.

This study had several limitations related to its retrospective nature. We were unable to verify the presence of pachydrusen in approximately 9% of PCV eyes due to extensive subretinal hemorrhage or exudate covering over four disc areas at the posterior pole. The assessed areas of OCT and fundus photography (FP) were not identical in size, such that the discordance might result in a poorer correlation between pachydrusen and pachyvessels than the previous results mentioned above. It was difficult to assess the correlation between pachydrusen near or outside the major retinal vessels due to image artifacts or lower resolution quality. The topographic association between pachydrusen and pachyvessels was not investigated in CSC eyes, to prevent statistical bias as the number of pachydrusen was low in CSC. Moreover, choroidal thickening resulting from diffusely dilated choroidal vessels within the fovea area in CSC makes it somewhat meaningless to run a subgroup analysis of pachydrusen and pachyvessels. The higher prevalence of diabetes and hypertension in the PCV group might be related to the older age of the PCV group, as an older age is a well-known risk factor for these systemic diseases [26,27]. This difference might lead to the significantly thinner subfoveal choroidal thickness in the PVC group compared to that of the CSC group, since age-related comorbidities such as hypertension and diabetes as well as age itself are known to be related to a decreased choroidal thickness [26,27]. However, this can also be considered as an inevitable limitation, based on the different age distribution by ocular disease entities.

In conclusion, the risk factors for pachydrusen differ by disease, even within pachychoroid spectrum diseases. Choriocapillaris attenuation due to a thickened Haller’s layer might be a common pathology for pachydrusen.

## 4. Materials and Methods

### 4.1. Patients

This study was approved by the Institutional Review Board of Ajou University Hospital, Suwon, Korea (IRB No.: AJIRB-MED-MDB-20-005), and it complied with the Declaration of Helsinki. The need for informed consent was waived by the Institutional Review Board of Ajou University Hospital given the retrospective nature of the study. Patients diagnosed with PCV and those with CSC at the Ophthalmology Department of Ajou University Hospital between January 2016 and December 2019 were included in this study. Demographic factors, such as age, sex, and systemic diseases, were retrospectively obtained from medical records.

PCV was primarily diagnosed by indocyanine green angiography (ICGA) (Spectralis HRA+OCT; Heidelberg Engineering, Heidelberg, Germany) findings showing single or multiple polypoidal aneurysmal bulges with or without a branching vascular network. CSC was diagnosed by fluorescein angiography (FA) and OCT findings showing serous neurosensory retinal detachment with or without idiopathic focal leaks. In CSC cases where the FA findings were not typical, we referred to ICGA findings presenting choroidal hyperpermeability with dilated choroidal vessels. When both eyes were diagnosed with either PCV or CSC, one eye was assigned to the PCV or CSC group, and the other eye was excluded from the contralateral group. The exclusion criteria were as follows: (1) history of vitrectomy, intravitreal injection, and/or laser photocoagulation; (2) those without OCT or FP data; (3) those with significant media opacities limiting the quality of imaging, (4) those with choroidal neovascularization, diffuse RPE atrophy, and other retinal diseases, including macular hole, epiretinal membrane, retinal vascular occlusion, and diabetic retinopathy, and (5) eyes with an extensive subretinal hemorrhage and/or exudate covering over four disc areas at the posterior pole in the FP that were considered as the “undetermined” group.

### 4.2. Imaging Analysis

Extracellular deposits were determined using FP and spectral domain OCT, based on the criteria used in previous studies [2,3]. Briefly, “pachydrusen” was identified by FP and OCT, defined as isolated or scattered yellowish white deposits in FP, and located under RPE [3]. Eyes without drusen or small drusen (<63 μm) were considered to have “no significant drusen”, and eyes with poorly visualized fundus due to media opacity were the “ungradable” group.

The location of the pachydrusen was determined as peripapillary (located within 1 disc diameter from the disc margin), subfoveal (located within 1 mm at the fovea), parafoveal (located within 3 mm at the fovea), perifoveal (located within 6 mm at the fovea), and along the vascular arcade (located within a 0.5 disc diameter from vessel) [25]. The distribution pattern was classified as follows: large solitary, clustered, and scattered [3]. Briefly, large solitary pachydrusen was defined by the presence of single pachydrusen with a size ≥ 125 μm; clustered pachydrusen by intermediate-sized pachydrusens (63 μm ≤ size < 125 μm) presented in clusters; and scattered pachydrusen by those of various sizes around optic disc and macular or along vascular arcade with spared macula [3]. The number of pachydrusen was counted manually using FP and OCT: large solitary and scattered pachydrusen were counted separately, while the number of clustered pachydrusen was considered as one if located within a 0.5 disc diameter. A dilated choroidal large vessel at Haller’s layer with attenuated overlying choriocapillaris (pachyvessel) was identified by OCT, and any association with pachydrusen was verified [3,21]. Any pigmentary change or pigment epithelium detachment was excluded. Representative cases of pachydrusen with pachyvessels are described in Figure 3.

The SFCT was measured as the vertical perpendicular distance from the innermost hyperreflective line of the chorioretinal interface to the hyperreflective line to the Bruch membrane in the enhanced depth imaging (EDI) mode of OCT. Considering that the mean SFCT in normal individuals is approximately 300 μm [28], those with SFCT ≥ 300 μm were classified as the “thick choroid” group and those with SFCT < 300 μm as the “thin choroid” group.

Color FPs were taken using an AFC-210 (NIDEK, Aichi, Japan) at a standard 45° centered on the fovea. The EDI mode was routinely obtained at the same time as the standard spectral domain OCT image (Heidelberg Spectralis OCT, Heidelberg Engineering, Heidelberg, Germany), and images were analyzed using Spectralis OCT software version 6.0. All color FP, OCT, FA, and ICGA findings were reviewed by two independent examiners (Y.H.K. and Y.-R.C.), and the agreement between the two examiners was good. In case of discrepancy, the third examiner (K.L.) analyzed the imaging findings. 

### 4.3. Statistical Analysis

All statistical analyses were performed using SPSS software (version 23.0; SPSS, Chicago, IL, USA). The baseline characteristics and pachydrusen-associated parameters were compared using an independent *t*-test, a Mann–Whitney U test, or a Kruskal–Wallis test for numerical values, and a chi-square test for categorical values. Logistic regression analysis was performed to verify the factors associated with the presence of pachydrusen. The McNemar test was performed to verify the association between included and fellow eyes for the presence of pachydrusen. The interobserver reliabilities were presented as the ICC for quantitative analyses. Statistical significance was set at *p* < 0.05.

## Figures and Tables

**Figure 1 ijms-23-08353-f001:**
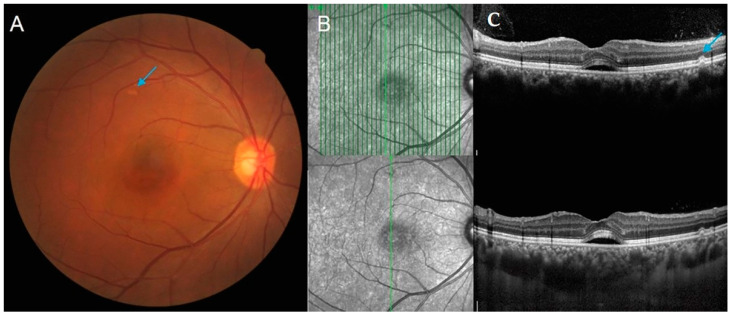
Representative cases of pachydrusen in CSC eye. (**A**) A 53-year-old male patient presented with CSC in his right eye, with a pachydrusen (blue arrows) in FP. (**B**) A solitary pachydrusen is noted superior to the fovea, and (**C**) thick subfoveal choroidal thickness (533 μm) with subretinal fluid is noted in the enhanced depth imaging mode. CSC = central serous chorioretinopathy; FP = fundus photography.

**Figure 2 ijms-23-08353-f002:**
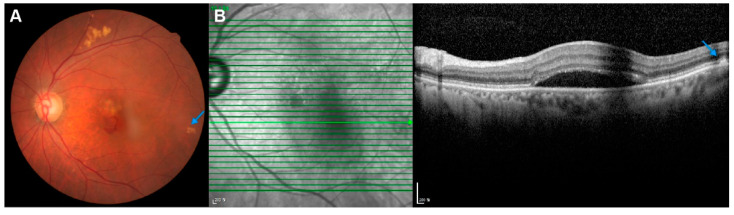
Representative cases of pachydrusen in PCV eye. (**A**) An 80-year-old male patient presented with subretinal hemorrhage due to PCV in his left eye, with a pachydrusen (blue arrows) at the temporal retina in FP. (**B**) Pachydrusen is noted at the temporal side in OCT. OCT = optical coherence tomography; PCV = polypoidal choroidal vasculopathy.

**Figure 3 ijms-23-08353-f003:**
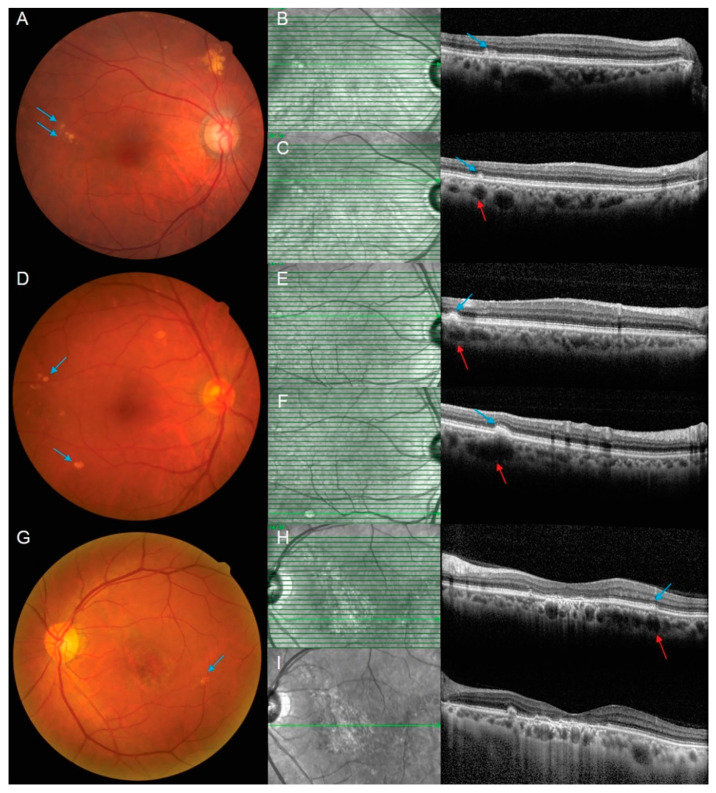
Representative cases of pachydrusen and pachyvessel. (**A**–**C**) An 80-year-old male patient diagnosed with PCV in his left eye presents with pachydrusen (blue arrows) in FP and pachydrusen and pachyvessel (red arrows) in OCT in the fellow eye. (**D**–**F**) A 69-year-old male patient diagnosed with PCV in his left eye presents with pachydrusen in FP and pachydrusen and pachyvessel in OCT in the fellow eye. (**G**–**I**) A 62-year-old male diagnosed with PCV in his left eye presents with pachydrusen in FP and pachydrusen and pachyvessel in OCT, as well as a relatively thin subfoveal choroidal thickness measured as 200 μm in the enhanced depth imaging mode. FP = fundus photography; OCT = optical coherence tomography; PCV = polypoidal choroidal vasculopathy.

**Table 1 ijms-23-08353-t001:** Baseline characteristics and OCT-related factors of included patients.

	PCV	CSC	*p* Value
No. of eyes	89	146	
Age	69.2 ± 8.5	45.3 ± 8.8	<0.001 ^†^
Sex, male	69 (78%)	116 (79%)	0.727
Diabetes	25 (28%)	8 (5%)	<0.001 *
Hypertension	50 (56%)	25 (17%)	<0.001 *
Mean SFCT (μm)	265.6 ± 105.1	402.3 ± 104.1	<0.001 ^†^
Presence of Pachydrusen	46 (52%)	29 (20%)	<0.001 *
No. of total pachydrusen	1.1 ± 1.6	0.3 ± 0.8	<0.001 ^†^
Pachydrusen subtype			
Large solitary	11 (12%)	4 (3%)	0.003 *
Clustered	28 (31%)	21 (14%)	0.001 *
Scattered	9 (10%)	6 (4%)	0.050

* *p* value < 0.05 by chi-square test. ^†^ *p* value < 0.05 by Mann–Whitney U test. CSC = central serous chorioretinopathy; PCV = polypoidal choroidal vasculopathy; SFCT = subfoveal choroidal thickness.

**Table 2 ijms-23-08353-t002:** OCT-related factors of included eyes with PCV stratified by age.

	Presence of Pachydrusen			No. of Pachydrusen		
	Thin Choroid	Thick Choroid	*p* Value	Thin Choroid	Thick Choroid	*p* Value
Total	20/55 (36%)	17/28 (61%)	0.035 *	0.8 ± 1.4	1.5 ± 1.7	0.022 ^†^
By age group (years)						
≤59 (n = 10)	1/7 (14%)	0/3 (0%)	1.000	0.2 ± 0.4	0	0.857
60–69 (n = 32)	6/18 (33%)	11/14 (79%)	0.016 *	0.8 ± 1.2	2.2 ± 2.0	0.022 ^†^
70–79 (n = 29)	9/19 (47%)	6/10 (60%)	0.700	0.9 ± 1.6	1.2 ± 1.2	0.377
≥80 (n = 12)	4/11 (36%)	0/1 (0%)	1.000	1.1 ± 1.7	0	0.667
*p* value within group	0.285	0.774		0.624	0.126	

* *p* value < 0.05 by chi-square test for the presence of pachydrusen. ^†^ *p* value < 0.05 by Mann–Whitney U test for the number.

**Table 3 ijms-23-08353-t003:** Presence of pachydrusens in PCV eyes according to location stratified by choroidal thickness.

Location of Pachydrusen	Thin Choroid	Thick Choroid	*p* Value
Peripapillae	2/46 (4%)	2/21 (10%)	0.584
Subfovea	0	0	N/A
Parafovea	1/46 (2%)	0	1.000
Perifovea	9/46 (20%)	9/21 (43%)	0.046 *
Vascular arcade	10/46 (22%)	5/21 (24%)	0.850

* *p* value < 0.05 by chi-square test for the presence of pachydrusen.

**Table 4 ijms-23-08353-t004:** Association of pachydrusen and pachyvessel in PCV eyes and fellow eyes.

Variables	PCV Eyes	Fellow Eyes
No. of total pachydrusen	82	101
No. of undetermined pachydrusen in OCT	35/82 (43%)	44/101 (44%)
No. of determined pachydrusen in OCT	47	57
Pachydrusen associated with pachyvessel	29/47 (62%)	38/57 (67%)
Pachydrusen without associated pachyvessel	18/47 (38%)	19/57 (33%)
Among PCV Eyes	Thin Choroid	Thick Choroid
No. of total pachydrusen	42	40
No. of undetermined pachydrusen in OCT	21 (50%)	14/40 (35%)
No. of determined pachydrusen in OCT	21	26
Pachydrusen associated with pachyvessel	13/21 (62%)	16/26 (62%)
Pachydrusen without associated pachyvessel	8/21 (38%)	10/26 (38%)

OCT = optical coherence tomography; PCV = polypoidal choroidal vasculopathy.

## Data Availability

The data presented in this study are available on request from the corresponding author.

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
