# Peer review of "The Association of Pachydrusen Characteristics with Choroidal Thickness and Patient’s Age in Polypoidal Choroidal Vasculopathy versus Central Serous Chorioretinopathy"

_ijms, 2022, doi:10.3390/ijms23158353_

Round 1
Reviewer 1 Report
The aim of the study is interesting, but the main limitation relies in the methods. The CSC population is poorly defined and the reader does not know if the analysed cases are of acute or chronic CSC. This influences the results.
Author Response
Thank you for your comment. Many authors use a basic distinction between acute CSC and chronic CSC based on the duration of subretinal fluid (SRF) and the structural changes visible on multimodal imaging. Although the serous detachment in acute CSC usually resolves within 3–4 months without the need for treatment, the detachment tends to persist in chronic CSC, and the chronic presence of SRF commonly leads to permanent structural damage in the neuroretina and RPE, with irreversible long-term vision loss. There are speculations that significant clinical differences exist between acute CSC and chronic CSC. However, acute CSC and chronic CSC share several genetic risk factors and possible pathophysiological overlap, particularly given similarities with respect to multimodal imaging [van Rijssen et al. Prog Retin Eye Res. 2019;73:100770]. In this respect, both of acute and chronic CSC present with the features of pachychoroidal spectrum diseases. Therefore, CSC population was not defined according to the duration of SRF. However, it should be noted that ICG and OCTA were further analyzed to leave out those with CNV or diffuse RPE atrophy cases.
Reviewer 2 Report
General Comments
Kim et al. in their research, quite comprehensively analyzed the issue of pachydrusen in pachychoroid spectrum diseases: polypoidal choroidal vasculopathy (PCV) and central serous chorioretinopathy (CSC). I believe that continuous progress in this area is required for a deeper understanding of the underlying mechanism of pachydrusen and its association with PCV and CSC. The literature on this topic is being continuously updated and hopefully, the presented work will be one of the needed clinical evidence from cohort studies. The work seems to be well-thought-out and well-organized, so I have no objections to the content and presentation of the results.
Author Response
We really appreciate this positive comment of the reviewer. Thank you.
Reviewer 3 Report
Thank you for giving me the chance to review your original study.
Although PCV and CSC both are pachychoroid spectrum diseases, they are different entities. Direct comparing variables between two entities seems non-sense.
There are pachychoroid spectrum diseases other than CSC, why choose CSC for pachydrusen comparison.
In the subgroup analysis of pachydrusen, only characteristics of PCV were analyzed instead of CSC.
Reviewer 4 Report
The authors investigated the relationship of pachydrusen with choroidal thickness and age in eyes with polypoidal choroidal vasculopathy (PCV), compared with eyes with central serous chorioretinopathy (CSC). This is an interesting study, although there are some discrepancies and ambiguities that should be addressed by the authors.
First, the title does not seem to be fully informative, because the authors assessed the association of pachydrusen characteristics with choroidal thickness and patient’s age depending on disease (PCV vs CSC) rather than the clinical characteristics of pachydrusen alone.
Abstract: The main focus of the study is not clearly defined in the abstract. First, the authors state that they assessed the relationship between pachydrusen and choroidal thickness and age in PCV eyes and fellow eyes vs CSC eyes. In line 15, they specifically mention the characteristics of pachydrusen, but except location, no results for these characteristics are reported. In the results, there is no mention of fellow eyes, and conclusions seem to have no link with the study objective or findings. The abstract should be edited for a more logical and coherent presentation of the background, objective, main findings, and conclusions of the study.
Lines 29-32: Note that there are more entities included under the umbrella term of pachychoroid spectrum disease. This should be stated more explicitly.
Line 109: Was the direct association between pachydrusen and pachyvessels investigated in PCV only? Why not CSC?
Lines 140-141: Interesting hypotheses on the pathogenesis of CSC have been put forward by Karska-Basta et al. Consider citing these studies to provide more background information and a wider context.
Izabella Karska-Basta, Weronika Pociej-Marciak, MichaÅ‚ ChrzÄ…szcz, Agnieszka Kubicka-TrzÄ…ska, Bożena Romanowska-Dixon, Marek Sanak: Altered plasma cytokine levels in acute and chronic central serous chorioretinopathy. Acta Ophthalmol., 2020; 23: 1–10. doi: 10.1111/aos.14547
Izabella Karska-Basta, Weronika Pociej-Marciak, MichaÅ‚ ChrzÄ…szcz, Joanna WilaÅ„ska, Martine J. Jäger, Anna Markiewicz, Bożena Romanowska-Dixon, Marek Sanak, Agnieszka Kubicka-TrzÄ…ska: Differences in anti-endothelial and anti-retinal antibody titers: implications for the pathophysiology of acute and chronic central serous chorioretinopathy. J. Physiol. Pharmacol., 2020; 71 (2): 1–8. doi: 10.26402/jpp.2020.2.07
Line 238: What ICGA device was used? Please specify.
Lines 239-240: Why the authors did not use ICGA for diagnosing CSC? Please explain.
Line 283: Please provide an interclass correlation between the readers.
105: The table title is unclear. Did you mean “presence of pachydrusen in included eyes with PCV” rather than “pachydrusen of included eyes”? Also, it seems that the table presents the results according to choroid thickness, but this is not reflected in the table title. Please revise.
Round 2
Reviewer 3 Report
Thank you for the explanation.
This manuscript is a resubmission of an earlier submission. The following is a list of the peer review reports and author responses from that submission.
Round 1
Reviewer 1 Report
Kim YH and their colleagues described clinical features of pachydrusen in PCV and CSC, and this retrospective study further compared the profiles between the diseases. However, most of the results have been already mentioned in previous reports (PCV: PMID29346242, 34314466, CSC: PMID33309963, 30852634). Therefore it is not clear that any new insights can be obtained from this study.
Reviewer 2 Report
Very interesting work comparing pachydrusen in two different disease entities.
However, because the integrity of the RPE-Bruch's membrane-Choroid is the epicenter of the pathology and its discussion in the paper, the CSC group should be carefully described due to the heterogeneity of the disease.
Line 168 "pachyvessel" written twice.